# Warming up human body by nanoporous metallized polyethylene textile

Lili Cai[1], Alex Y. Song [2], Peilin Wu[1], Po-Chun Hsu[1], Yucan Peng[1], Jun Chen[1], Chong Liu[1], Peter B. Catrysse[2], Yayuan Liu [1], Ankun Yang[1], Chenxing Zhou[1], Chenyu Zhou[1], Shanhui Fan[2] & Yi Cui[1,3]

Space heating accounts for the largest energy end-use of buildings that imposes significant burden on the society. The energy wasted for heating the empty space of the entire building can be saved by passively heating the immediate environment around the human body. Here, we demonstrate a nanophotonic structure textile with tailored infrared (IR) property for passive personal heating using nanoporous metallized polyethylene. By constructing an IR-reflective layer on an IR-transparent layer with embedded nanopores, the nanoporous metallized polyethylene textile achieves a minimal IR emissivity (10.1%) on the outer surface that effectively suppresses heat radiation loss without sacrificing wearing comfort. This enables 7.1 °C decrease of the set-point compared to normal textile, greatly outperforming other radiative heating textiles by more than 3 °C. This large set-point expansion can save more than 35% of building heating energy in a cost-effective way, and ultimately contribute to the relief of global energy and climate issues.

[1] Department of Materials Science and Engineering, Stanford University, Stanford, CA 94305, USA. [2] E.L. Ginzton Laboratory, Department of Electrical Engineering, Stanford University, Stanford, CA 94305, USA. [3] Stanford Institute for Materials and Energy Sciences, SLAC National Accelerator Laboratory, 2575 Sand Hill Road, Menlo Park, CA 94025, USA. Correspondence and requests for materials should be addressed to Y.C. (email: yicui@stanford.edu)

Indoor heating and cooling in residential and commercial buildings take up an enormous portion of the total energy consumption worldwide. In the US, for instance, the buildings sector accounts for about 41% of primary energy consumption, with space heating and cooling making up about 37% of that[1, 2]. The need for heating or cooling the vast empty space of the entire building to maintain thermal comfort of the human body leads to substantial energy waste and significantly contributes to the energy crisis and global warming[3, 4]. The passive personal thermal management strategy that provides localized thermal control of the immediate environment around the human body to mitigate the energy demand for indoor temperature regulation could therefore offer a promising low-cost complement to efforts that focus on thermal exchange of the entire building for energy savings[5–12]. Studies show that an expansion of the cooling and heating set-points by 4 °C each can lead to large energy savings up to 45 and 35%, respectively[13].

For indoor environment, where most people stay in a sedentary state, more than 50% of the heat generated by the human body is dissipated through infrared (IR) radiation[14, 15]. Therefore, the control of IR radiation by tailoring the thermal radiation properties of textiles can have a significant impact on localized heating or cooling of the human body under indoor conditions. However, traditional textiles only act as an insulation layer against thermal conduction and convection, but lack the control of thermal radiation. Our recent work demonstrated effective radiative cooling of the human body by 2 °C using IR-transparent nanoporous polyethylene (nanoPE) textile, which shows the great potential of engineering the thermal radiation properties of textiles for energy savings in buildings[5].

Compared to cooling, much larger energy savings can be expected by developing radiative heating textiles, as space heating (22.5%) holds much larger proportion of all the energy consumed in the buildings sector than space cooling (14.8%)[2]. However, there is lack of a desirable radiative heating textile that has both the optimal heating capability and good wearability. The very few existing radiative textiles all have severe deficiencies. For example, the Mylar blanket, consisting of a solid plastic sheet (often polyethylene terephthalate (PET) film) that is coated with a dense metallic reflecting film, lacks breathability, making it uncomfortable for everyday wear[16]. The Omni-Heat technology, which prints sparse metallic dots onto the inside of garments to reflect human body heat, suffers from low reflectivity and poor radiative heating performance[17]. The recently developed Ag nanowire (AgNW)-coated textile has low IR reflection of ~40% that only warms up the human body by 0.9 °C higher than the normal textiles[7]. These deficiencies can be mainly attributed to two factors. First, all the existing radiative heating textiles are based on the concept of reflecting back the human body IR radiation to reduce heat loss, which is in fact not the most effective IR radiation control approach for localized human body heating. A fundamental understanding on the heat transfer model is needed to provide the theoretical guidance. Second, there is always a dilemma between optimal radiative heating performance and good wearability, which needs to be addressed by an advanced photonic structure design.

Here, we report a nanophotonic structure textile with tailored IR property that shows superior passive radiative personal heating capability ever achieved with no sacrifice of wearability. To bridge the gap in theoretical understanding of the required spectral properties for heating textiles, we perform the first heat transfer model analysis for personal heating, which reveals the necessity of low-IR emissivity on the textile outer surface, instead of high IR reflectivity on the inner surface, in suppressing the radiative heat dissipation of clothed human. Furthermore, structural photonic simulation indicates that interconnected nanoporous metallic film has almost 100% IR reflectivity as a solid film, which can thus solve the dilemma between optimal thermal radiation property and good breathability. On the basis of the design guidance from thermal and photonic simulations, we experimentally develop an advanced nanophotonic structure textile by constructing an IR-reflective metallic layer on an IR-transparent PE layer with embedded nanopores that are smaller than the IR wavelength but larger than the water molecule. The resulting nanoporous metallized PE textile achieves a minimal IR emissivity of 10.1% on the non-metallized PE side, and in the meantime maintains good wearability as normal textiles. Thermal measurements show that the nanoporous metallized PE textile, when laminated on the outer surface of traditional textiles rather than placed on the inside, enables 7.1 °C decrease in the set-point of ambient temperature compared to traditional cotton textile, greatly outperforming all the existing radiative heating textiles, including AgNW-coated textile (0.9 °C), Omni-Heat (0.1 °C) and Mylar blanket (4 °C). We believe the superior localized heating capability of this wearable nanoporous metallized PE textile could potentially bring forth significant impact on reducing global energy consumption.

## Results

**Heat transfer model analysis.** In previous studies, heat transfer model analysis was developed to evaluate the impact of textile's IR properties on personal cooling[5, 6]. Such theoretical analysis has not yet been applied to personal heating. In this regard, we first performed one-dimensional steady-state heat transfer model analysis of clothed human skin to determine the required IR properties of textiles for maintaining thermal comfort in cold environment (Supplementary Fig. 1). This can serve as a guidance in designing radiative heating textiles. In this model, we assume constant skin temperature (33 °C) and heat generation rate (73 W m$^{-2}$), and set the criterion of thermal comfort as the equality of the total heat dissipation rate with the total heat generation rate. Under these conditions, the set-point of ambient temperature for maintaining thermal comfort is a function of the IR properties of the textiles (see Supplementary Note 1 and Supplementary Table 1 for more details).

Figure 1a plots the calculated set-points of ambient temperature for textiles with varying emissivity at the inner ($\varepsilon_i$) and outer surfaces ($\varepsilon_o$). The set-point decreases monotonically with decreasing $\varepsilon_o$ at fixed $\varepsilon_i$, while it remains almost constant when $\varepsilon_i$ is changed at fixed $\varepsilon_o$. These results indicate the decisive role of the textile's outer surface IR emissivity in controlling the textile's heating performance, which is contradictory to the common belief that a reflective inner surface is required to reflect back the IR radiation from the human body to keep warm. The little impact of the inner surface IR reflectivity on heating is mainly attributed to the dominancy of heat conduction over radiation in the heat transfer between the skin and the textile inner surface, where the conductive thermal resistance is quite small for typical fitting tightness. In the heat dissipation from the textile outer surface to the environment, heat radiation can contribute more than 50% for textiles with high IR emissivity under indoor conditions where the convective transfer coefficient is typically low. Thus, for the goal of decreasing the heating set-point, it is crucial to reduce the textile's outer surface IR emissivity, which allows the suppression of radiative heat dissipation from a clothed human body to the ambient environment. Normal textiles like cotton (Fig. 1b and Supplementary Table 2), however, lack the desired radiation control due to their high emissivity of 0.8 ~ 0.9, and a set-point of about 22 °C is needed for maintaining thermal comfort. For an ideal radiative heating textile that has no IR emission at the outer surface, a minimal set-point of about 12 °C can be reached according to the calculation in Fig. 1a.

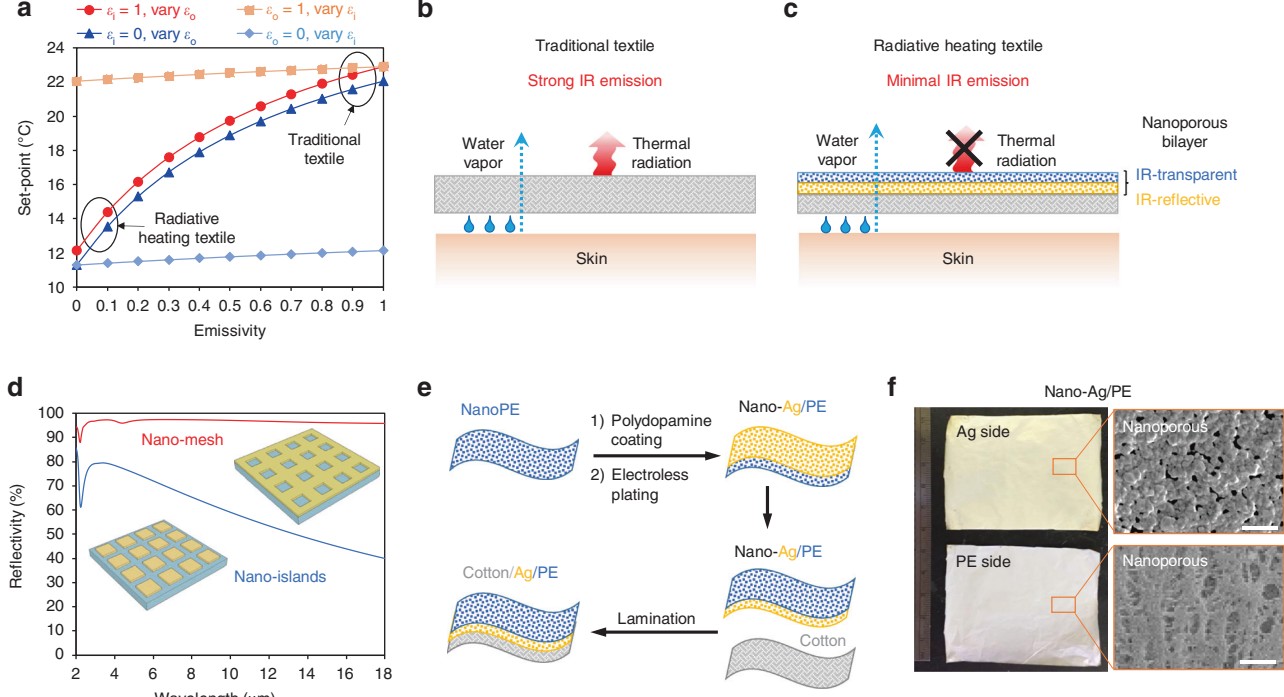

**Fig. 1** Simulation and fabrication of radiative heating textile. **a** Simulated set-points of ambient temperature for maintaining thermal comfort as a function of the textile's IR emissivity at the inner surface ($\varepsilon_i$) and outer surface ($\varepsilon_o$). Schematics depicting the heat dissipation and vapor transmission of human body covered with **b** traditional textile and **c** nanophotonic structure heating textile composed of an IR-transparent layer and an IR-reflective layer with embedded nanopores in both layers to simultaneously achieve minimal IR emissivity and good breathability. **d** Simulated total IR reflectance of metallic nano-island film and metallic nano-mesh film with coverage density of 81%. The island size is 900 nm × 900 nm. The hole size is 440 × 440 nm. For both cases, the period is 1 µm × 1 µm. **e** Schematic representation of the fabrication process of nano-Ag/PE textile, which is laminated on the outer surface of cotton. **f** Photos and SEM images of the Ag side and PE side of nano-Ag/PE. *Scale bar*, 1 µm

**Nanophotonic structure heating textile design**. Based on the thermal radiation relation $\varepsilon = 1 - \rho - \tau$ ($\varepsilon$, $\rho$ and $\tau$ are emissivity, reflectivity and transmissivity, respectively), we propose a bilayer nanophotonic structure textile composed of an IR-reflective metallic layer and an IR-transparent PE layer with embedded nanopores in both layers to simultaneously have minimal IR emissivity and good breathability (Fig. 1c). In this nanoporous metallized PE design, the embedded nanopores in the metallic layer are smaller than the IR wavelength but larger than the water molecule, which overcomes the limitations of other radiative heating textiles where the metallic coating is either too dense to be breathable (Mylar blanket, Supplementary Fig. 2a) or too sparse to be highly reflective (Omni-Heat, Supplementary Fig. 2b). Although the previously reported AgNW coating formed the desired nanoscale pores in the size of 200 ~ 300 nm, its reflectivity is still far below the simulated value[7]. This is probably due to the imperfect mutual connection within the AgNW network on the wavy surface of textiles. Our rigorous coupled-wave analysis shows that nanoporous metallic film in the form of an interconnected mesh has far higher reflectivity than that in the form of isolated islands with the same coverage density, approaching IR reflectivity of 100% as a solid film (Fig. 1d). Therefore, it is essentially important to attain a well interconnected nanoporous mesh for the metallic layer to simultaneously render optimal reflectivity and breathability.

On the other hand, a key superiority of this design is the use of nanoPE as the support/protection layer for the metallic layer. In addition to permitting air permeability, the nanopores in the size range comparable with the wavelength of visible light (400–700 nm) can scatter visible light strongly and make PE opaque to human eyes. While the pore sizes are still much smaller

than the IR wavelength (~ 9 µm), the nanoPE film is still highly transparent to IR. Therefore, the nanoPE layer can serve as the outer support/protection layer covering the metallic layer to avoid exposure of metal for better durability and appearance, while ensuring the minimal IR emission in conjunction with the metallic layer. As a result, by laminating the nanoporous metallized PE textile on the outside of traditional textile, its minimal IR emissivity can effectively suppress the radiative heat dissipation of the human body to the environment without blocking the transmission of water vapor. The inner traditional textile in contact with the skin can maintain wearing comfort and insulation against thermal conduction.

**Fabrication and IR property of nanoporous metallized PE**. The nanoporous metallized PE textile was fabricated by coating nanoporous Ag film on 12-µm-thick nanoPE (nano-Ag/PE) via a low-cost and scalable electroless plating method[18], which can be then laminated onto the outside of cotton with nanoPE side facing the ambient environment (cotton/Ag/PE), as shown in Fig. 1e (see Methods for more details). Prior to the electroless plating, the nanoPE was treated with polydopamine (PDA) which modifies its surface from being hydrophobic to hydrophilic and enhances its adhesion with the coated Ag film[19–23]. With the optimized plating time, the coated Ag film shows nanoscale pores in the size of 50–300 nm (Fig. 1f). These nanopores in the Ag film and the interconnected pores in the nanoPE (Fig. 1f, 50–1000 nm in size) both provide pathway for the transmission of water vapor, ensuring good breathability of this nano-Ag/PE shell layer.

The total IR reflectance of the coated nanoporous Ag film was measured using a Fourier transform infrared spectrometer (FTIR)

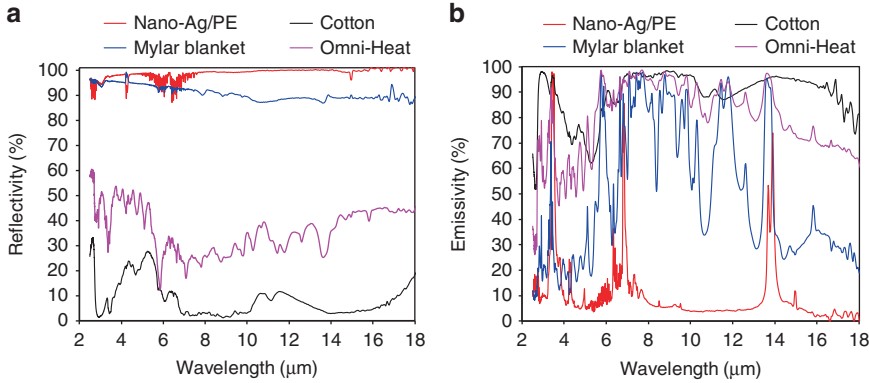

**Fig. 2** Infrared property characterizations. **a** Measured total FTIR reflectance of the metallic coating side of nano-Ag/PE, cotton, Mylar blanket and Omni-Heat. **b** Measured total FTIR emittance at the textile outer surface of nano-Ag/PE, cotton, Mylar blanket and Omni-Heat

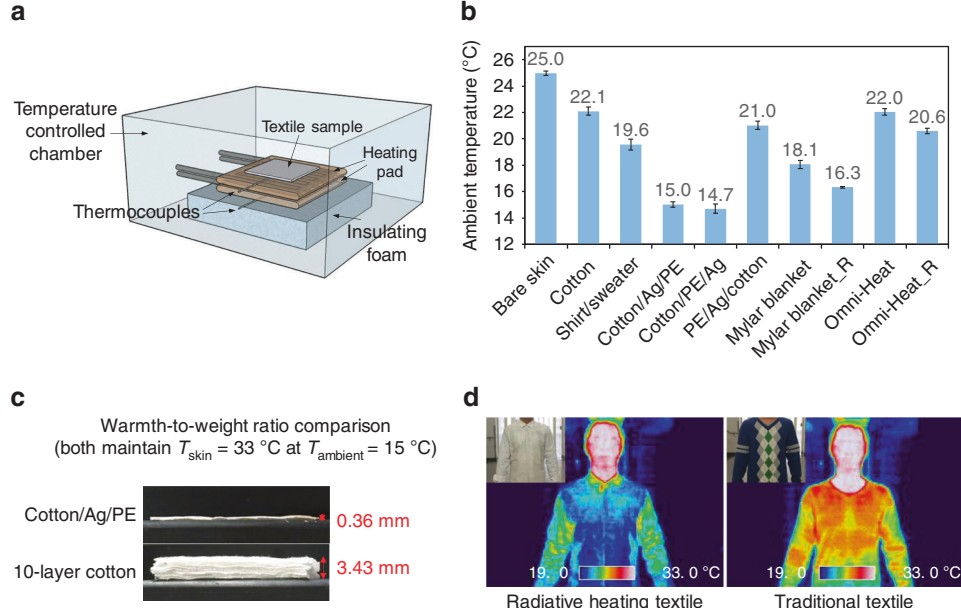

**Fig. 3** Thermal performance evaluations. **a** Schematic representation of the thermal measurement set-up, which measures the required set-point of ambient temperature for a textile to maintain constant skin temperature at constant heat generation rate. **b** Measured set-points of ambient temperature for bare skin and skin covered with textile samples. In the notation of cotton/Ag/PE, cotton/PE/Ag and PE/Ag/cotton, the layer on the left is in contact with skin and the layer on the right is facing ambient environment. Mylar blanket_R and Omni-Heat_R correspond to the reversed Mylar blanket and Omni-Heat, respectively. In the reversed case, the metallic side is facing outside. For the measurements of Mylar blanket and Mylar blanket_R, a cotton layer was added below them to ensure that they have the same conductive insulation as the other tested textiles. **c** Comparison of warmth-to-weight ratio between cotton/Ag/PE and cotton. **d** Thermal imaging and photos (insets) of human body wearing garments made from radiative heating cotton/Ag/PE textile and traditional textile, respectively. All *error bars* represent the standard deviation

equipped with a diffuse gold integrating sphere (Fig. 2a). The coated nanoporous Ag film shows a high IR reflectivity of 98.5% (weighted over human body radiation), which is much higher than those of the normal textile like cotton (8.4%), the metallic dots of Omni-Heat (31.9%) and the dense metal film of Mylar blanket (91.2%). Such high IR reflectivity of the nanoporous Ag film is consistent with the structural photonic simulation results. We further measured the IR emissivity on the outer nanoPE side of the nano-Ag/PE textile (Fig. 2b; the reflectance and transmittance are shown in Supplementary Fig. 3). Owing to the high IR reflectivity of the nanoporous Ag film (98.5%) and the high IR transparency of nanoPE (96.0%), the nano-Ag/PE textile shows a low IR emissivity of 10.1% on the outer surface, which is the lowest comparing to Mylar blanket (60.6%), Omni-Heat (85.4%) and cotton (89.5%). Note that the IR emissivity on the plastic sheet side of Mylar blanket (60.6%) is much higher than

that on the nanoPE side of cotton/Ag/PE (10.1%), despite their comparable IR reflectivity of the metallic coating. This result illustrates the importance of high IR transparency for the outer protection layer in order to retain the low IR emissivity of the underlying metal layer.

**Passive radiative heating performance**. We evaluated the heating performance of the laminated cotton/Ag/PE textile using a device that simulates the heat generation of skin inside a temperature controlled chamber (Fig. 3a). This set-up allows the measurement of the required set-point of ambient temperature for maintaining the skin temperature at 33 °C when covered with different textiles (Fig. 3b). Lower set-point corresponds to better heating performance of a textile. For the cotton/Ag/PE textile, the required set-point is measured as 15.0 °C, which is 7.1 °C lower

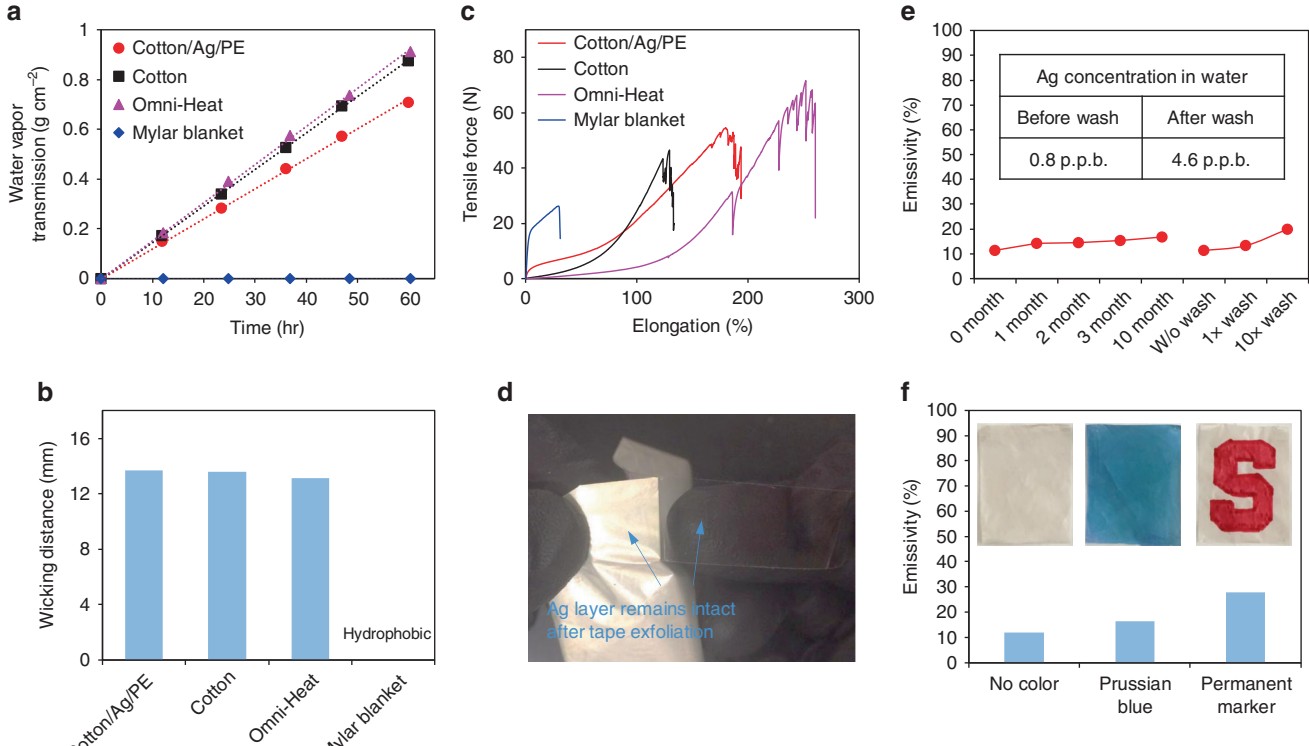

**Fig. 4** Wearability tests. Comparison of **a** water vapor transmission rate, **b** wicking distance and **c** tensile strength for cotton/Ag/PE, cotton, Omni-Heat and Mylar blanket. **d** Adhesion test shows that the coated Ag film is strongly bonded with PE and cannot be removed by an adhesive tape. **e** Durability test of the cotton/Ag/PE textile sample against washing and handling over time, showing little degradation of the IR emissivity. Inset table lists the Ag concentration in water before and after washing, confirming that negligible amount of Ag is released from the textile sample during washing. **f** IR emissivity comparison of the textile samples with and without coloration. Inset shows photos of the uncolored and colored textiles

than that of cotton (22.1 °C). The thermal performance of the cotton/Ag/PE textile is also better than typical indoor winter garments, e.g., a shirt and a sweater, which requires a set-point of 19.6 °C. For normal textile such as cotton that only has insulation against thermal conduction and convection, it needs bulky thickness of 10 layers to reach the same insulation performance as one layer of cotton/Ag/PE with thermal radiation control (Fig. 3c). This warmth-to-weight ratio comparison clearly illustrates the light-weight advantage of low-IR-emissivity textile over traditional textile for localized human body heating. Furthermore, the radiative heating property of the cotton/Ag/PE textile is validated by wearing on the human body and visualized using thermal imaging (Fig. 3d). In thermal imaging, colder color indicates less emission of radiative heat. The color comparison here illustrates that the human body emits minimal IR heat to keep warm when wearing a garment made from cotton/Ag/PE textile (colder color), while traditional textile emits IR heat to the environment substantially (warmer color).

Compared to the commercially available Omni-Heat and Mylar blanket, our cotton/Ag/PE textile also shows much better heating performance as a result of the nanophotonic structure design. First, the low-IR-emissive nano-Ag/PE textile is laminated on the outside of cotton to most effectively suppress the radiative heat dissipation to the environment. In comparison, when the nano-Ag/PE shell is put on the inside of cotton (PE/Ag/cotton), the measured set-point of 21.0 °C is about the same as that for cotton (22.1 °C), confirming the little effect of the inner surface emissivity/reflectivity on radiative thermal insulation. This also explains the fact that Omni-Heat with metallic coating on the inside shows the same set-point (22.0 °C) as cotton (22.1 °C), whereas the reversed Omni-Heat (Omni-Heat_R) with metallic

coating on the outside shows a lower set-point of 20.6 °C. The second advantageous design in cotton/Ag/PE is the use of IR-transparent nanoPE as the protection layer, which can retain the low-IR emissivity of the underlying metallic surface. As a result, the cotton/Ag/PE textile avoids the exposure of metal on the textile surface for the reason of better durability and appearance, and still achieves a low set-point of 15.0 °C that is comparable to cotton/PE/Ag with the metallic layer facing outside (14.7 °C). In contrast, Mylar blanket with the IR-absorbing/emitting PET film facing outwards requires higher set-point (18.1 °C) than that of reversed Mylar blanket (Mylar blanket_R) with metallic coating on the outside (16.3 °C). Finally, their thermal radiation properties are visualized using thermal imaging (Supplementary Fig. 4), confirming the thermal measurement results in Fig. 3b.

**Wearability evaluations.** Besides the heating performance, we further tested the wearability of the cotton/Ag/PE textile. First, the cotton/Ag/PE textile has water vapor transmission rate at ~ 0.012 g cm$^{-2}$ h$^{-1}$, comparable to that of cotton and Omni-Heat at ~ 0.015 g cm$^{-2}$ h$^{-1}$ (Fig. 4a). This demonstrates the nanopores in the Ag/PE shell are permeable enough for transmitting water vapor from perspiration by natural diffusion and convection. In contrast, Mylar blanket with nonporous plastic film and dense metal coating is completely non-permeable. Second, owing to the laminated cotton layer, the cotton/Ag/PE textile shows comparable wicking rate (~ 1.3 cm s$^{-1}$) and mechanical strength (maximum endurable tensile force of 50 N) as cotton and Omni-Heat (Figs. 4b and c). Third, we demonstrated that the electrolessly plated Ag layer on nanoPE remains intact even after

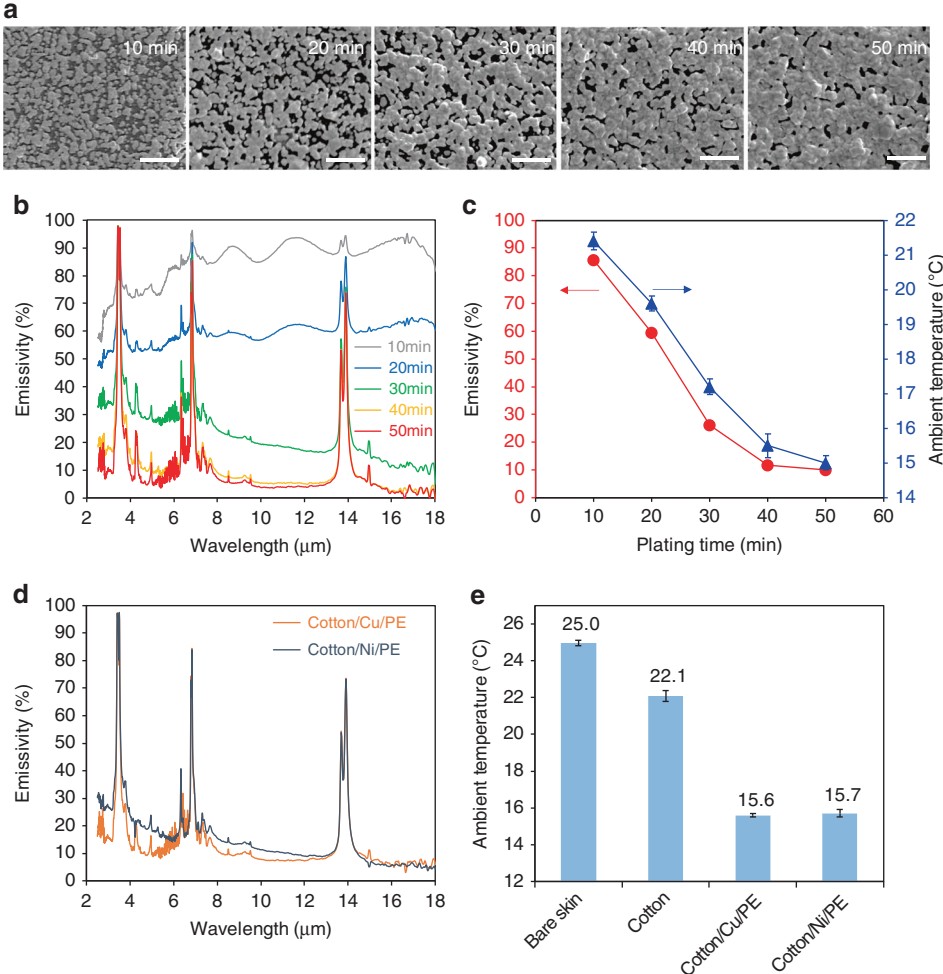

**Fig. 5** Tunability of IR radiation property and versatile choices of low-cost metals. **a** SEM images of the electrolessly plated Ag film with varying plating time from 10 to 50 min. *Scale bar*, 1 μm. **b** IR emittance spectra of the cotton/Ag/PE textiles with different Ag plating time. **c** Plots of the average IR emissivity (normalized over human body radiation) and the measured set-point of ambient temperature vs. the Ag plating time. **d** Measured IR emittance of the cotton/Cu/PE and cotton/Ni/PE textile samples. **e** Comparison of the set-points for bare skin, cotton, cotton/Cu/PE and cotton/Ni/PE. All *error bars* represent the standard deviation

the peeling by an adhesive tape, showing the strong adhesion of the Ag film with nanoPE enabled by the adhesive PDA coating (Fig. 4d). With such strong adhesion, the cotton/Ag/PE textile shows great durability against washing and handling over time with little degradation of the IR emissivity (Fig. 4e). We also measured the Ag concentration in water before and after washing the cotton/Ag/PE textile in swirling clean water using inductively coupled plasma mass spectrometry. The Ag concentration after wash slightly increases from 0.8 to 4.6 p.p.b., but still remains at p.p.b. level, which confirms that negligible amount of Ag is released from the cotton/Ag/PE textile during washing. In addition, due to the colorable PE protection layer on the outer surface, the cotton/Ag/PE textile can be colored as normal textile to enhance its appearance for better wearability. We tested the IR emissivity of the cotton/Ag/PE textile after being colored with Prussian blue and permanent marker dye (Supplementary Fig. 5). It is found that low-IR-absorbing pigment like Prussian blue results in smaller increase of IR emissivity of the textile than high-IR-absorbing pigment like permanent marker dye (Fig. 4f). Therefore, it is critical to choose pigments with low IR absorption for coloring the radiative heating textiles.

Furthermore, the ability to tailor the radiative heating capability of a thermal textile is important for the adaptation to different weather conditions. By varying the electroless plating time, the Ag coverage density of our cotton/Ag/PE textile can be tuned (Fig. 5a). This can lead to textiles with tailored IR emissivity (Fig. 5b) that require different set-points for maintaining thermal comfort (Fig. 5c). This tunability is quite useful in making suitable garments for variable levels of coldness in different seasons. Importantly, unlike conventional textiles such as cotton, where enhanced thermal insulation comes from the use of a thicker textile, here the tunability is achieved without significant increase of weight or thickness of the textile. Finally, it is worthwhile to note that the amount of Ag needed for the low-IR-emissivity textile is quite small. As shown in Supplementary Fig. 6, the thickness of the coated Ag film is only ~150 nm, so it only needs 2.8 g of Ag that costs $1.5 to make a garment covering the whole surface area of the human body (~ 1.8 m²). Besides, the cost can be further reduced by replacing Ag with other more cost-effective metals, such as copper (Cu) and nickel (Ni). As shown in Figs. 5d and e, both cotton/Cu/PE and cotton/Ni/PE demonstrate low IR emissivity and require low set-points for maintaining thermal comfort. With proper passivation to ensure their durability against oxidation, these nonprecious metals can serve as more cost-effective options.

## Discussion

In summary, we demonstrated a nanophotonic structure textile using nanoporous metallized PE as the outer shell to suppress radiative heat dissipation for localized human body heating. This textile design achieves a nanoporous metallic coating with high IR reflectivity of 98.5% on IR-transparent nanoPE (96.0%), leading to a low IR emissivity of 10.1% on the textile outer surface. Such a low-IR-emissivity textile enables 7.1 °C decrease in the set-point of ambient temperature compared to traditional cotton textile, which is the best heating performance ever achieved by all the existing radiative heating textiles. The 7.1 °C set-point reduction can potentially result in building heating energy savings of more than 35%[13]. Moreover, wearability tests showed that this textile is light-weight, breathable, durable, washable and colorable as normal textiles. We believe the large set-point expansion enabled by this superior radiative heating textile can greatly mitigate the energy demand for indoor heating, and ultimately contribute to the relief of global energy and climate issues.

## Methods

**Textile fabrication**. The nanophotonic structure textile was fabricated by electrolessly plating nanoporous metallic film onto nanoPE and then laminating the metallized nanoPE with cotton. Before electroless plating, the nanoPE surface was modified with polydopamine (PDA) coating for 2 h in an aqueous solution that consists of 2 g l$^{-1}$ dopamine hydrochloride (Sigma Aldrich) and 10 mM Tris-buffer solution (pH 8.5, Teknova)[19]. For electroless plating of silver (Ag), the PDA-coated nanoPE was first dipped into a 25 g l$^{-1}$ AgNO$_3$ solution (99.9%, Alfa Aesar) for 30 min to form the Ag seed layer. The seeded nanoPE was thoroughly rinsed with deionized (DI) water, and then immersed into the plating bath solution containing 4.2 g l$^{-1}$ Ag(NH$_3$)$_2$$^+$ (made by adding 28% NH$_3$·H$_2$O dropwise into 5 g l$^{-1}$ AgNO$_3$ until the solution became clear again) and 5 g l$^{-1}$ glucose (anhydrous, EMD Millipore Chemicals)[18]. For electroless plating of copper (Cu) and nickel (Ni), the PDA-coated nanoPE was first pretreated by sensitizing in SnCl$_2$ (5 g l$^{-1}$ SnCl$_2$ with 10 ml l$^{-1}$ HCl, Sigma Aldrich) and catalyzing in PdCl$_2$ (0.5 g l$^{-1}$ PdCl$_2$ with 6.25 ml l$^{-1}$ HCl, Sigma Aldrich)[24]. After rinsing in DI water, the catalyzed nanoPE was then immersed into the respective plating bath to coat Cu or Ni film. The Cu plating bath contains 0.05 M CuSO$_4$ (99%, Sigma Aldrich), 0.05 M tetrasodium ethylenediaminetetraacetate (Na$_4$EDTA, 99%, Sigma Aldrich), 0.1 M boric acid (99.9%, Mallinckrodt Chemicals) and 0.1 M dimethylamine borane with pH adjusted to ~ 8 by NaOH (DMAB, 97%, Sigma Aldrich). The Ni plating bath contains 20 g l$^{-1}$ NiSO$_4$ (99%, Sigma Aldrich), 10 g l$^{-1}$ sodium citrate (99%, Sigma Aldrich), 5 g l$^{-1}$ lactic acid (85%, Sigma Aldrich), 5.5 g l$^{-1}$ DMAB and 11 ml l$^{-1}$ NH$_3$·H$_2$O (28%, EMD Millipore Chemicals).

**Material characterization**. The IR reflectivity ($\rho$) and transmittance ($\tau$) were measured using a FTIR spectrometer (Model 6700, Thermo Scientific) accompanied with a diffuse gold integrating sphere (PIKE Technologies). The IR emissivity ($\varepsilon$) was then calculated using equation $\varepsilon = 100\% - \rho - \tau$. The scanning electron microscope (SEM) images were taken by FEI Sirion (5 kV).

**Thermal measurement**. The skin is simulated by a silicone rubber fiberglass insulated flexible heater (Omega, 72 cm$^2$) that is connected to a power supply (Keithley 2400). A ribbon type hot junction thermocouple (0.3 mm in diameter, K-type, Omega) is in contact with the top surface of the simulated skin to measure the skin temperature. A guard heater and an insulating foam are placed below the simulated skin heater to ensure that the heat generated by the skin heater only transfer to the ambient. The temperature of the guard heater is always set the same as the skin heater, so downward heat conduction to the table is averted. The whole device is enclosed in a chamber, and the ambient temperature inside the chamber can be controlled. We set the power density of the skin heater to be constant at 73 W m$^{-2}$, which renders the skin temperature of 33 °C at the ambient temperature of 25 °C. When the skin was covered by the textile sample (5 × 7 cm$^2$), we measured the steady-state ambient temperature that is needed to maintain the skin temperature at 33.0 ± 0.1 °C. The thermal images were taken by a calibrated thermal camera (MikroSHOT, Mikron).

**Water vapor transmission rate test**. This test procedure is based on ASTM E96 with modification. 100 ml media bottles (Fisher Scientific), filled with 60 ml distilled water, were sealed by the textile samples using open-top caps and silicone gaskets (Corning). The sealed bottles were then placed into an environmental chamber. The temperature and relative humidity inside the chamber were held at 35 °C and 30 ± 10%, respectively. The total mass of the bottles together with the samples was measured periodically. The reduced mass, corresponding to the evaporated water, was then divided by the exposed area (3 cm in diameter) to derive the water vapor transmission rate.

**Wicking test**. This test procedure is based on AATCC TM 197 with modification. The textile samples were cut into 2-cm-wide strips and dipped into distilled water. The water climbs up the sample due to capillary force. The climbing distance in the duration of 10 s was measured for the textile samples.

**Mechanical test**. The tensile strength test was measured by Instron 5565. The textile samples were cut into the shape of 2 cm wide and 5 cm long. The gauge distance was 3 cm long, and the displacement rate was kept at 10 mm min$^{-1}$.

**Washing test**. The cotton/Ag/PE textile (5 cm × 7 cm) was washed in clean water (100 ml) under stirring for 12 h. The water before and after wash was collected, and then tested using inductively coupled plasma mass spectrometry (ICP-MS) to quantify the amount of Ag released from the textile sample during washing.

**Coloration**. For coloration with Prussian blue, the PDA-coated nanoPE was first immersed in 0.16 M FeCl$_3$ (97%, Alfa Aesar) solution (15 ml) for 20 min. The functional groups in PDA can reduce Fe$^{3+}$ to Fe$^{2+}$, and bind the reduced Fe$^{2+}$ ions on the nanoPE surface. Next, 0.12 M K$_3$Fe(CN)$_6$ (99%, ACROS Organics) with the same volume was added dropwise into the above solution. The [Fe(CN)$_6$]$^{3-}$ ions then react with the Fe$^{2+}$ ions on the nanoPE surface to form Prussian blue (Fe$^{III}$$_4$[Fe$^{II}$(CN)$_6$]$_3$·$x$H$_2$O), resulting in blue color. For coloration with permanent maker dye, a permanent maker pen (Sharpie) in red color was used to directly paint on the textile surface.

**IR reflectance simulation**. We employed rigorous coupled-wave analysis method to simulate the reflection of IR light from the metal coating[25]. We modeled the metal coating in different structures, including disconnected island film and interconnected mesh film. Both films have thickness of 100 nm and area coverage of 81%. The island size is 900 nm × 900 nm. The hole size is 440 × 440 nm. For both cases, the period is 1 µm × 1 µm. Computation code is available from the authors upon request.

**Data availability**. The data sets generated during and/or analyzed during the current study are available from the corresponding author on reasonable request.

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

## Acknowledgements

This work was sponsored by the Advanced Research Projects Agency-Energy (ARPA-E), the US Department of Energy, under Award Number DE-AR0000533. The authors acknowledge Prof. Hongjie Dai for lending the thermal camera.

## Author contributions

Y.C. and L.C. conceived the idea. L.C. carried out electroless plating of silver, SEM and FTIR characterizations, thermal measurements and wearability tests. L.C. and P.B.C. conducted heat transfer model analysis. A.Y.S. performed the rigorous coupled-wave simulation. P.W. carried out electroless plating of copper and nickel. P.-C.H. provided discussion of the experiments and results. L.C., Y.P. and J.C. performed tensile strength measurement. C.L. performed ICP-MS. Y.L. helped with paper revision. L.C. and A.Y. carried out thermal imaging. C.X.Z. and C.Y.Z. assisted in FTIR and thermal measurements. Y.C. and S.F. supervised the project. L.C. and Y.C. wrote the paper. All authors discussed the results and commented on the manuscript.

## Additional information

**Competing interests:** The authors declare no competing financial interests.

