## [Peer Review File · Nature Communications]

Reviewers' comments:

Reviewer #1 (Remarks to the Author):

In this paper, the authors reported the application of the nanoporous metallized PE as the outer shell to suppress radiative heat dissipation for localized personal heating. The heat transfer, IR property and wearability were fully investigated. However, the core concept, tailoring the infrared radiate property by structure design (such as altering nano pore diameter and fiber diameter) and constructing metallized reflect layer, is ordinary due to lack of novelty. As far as I know, there have been several studies published about the similar idea (ACS Photonics, 2015,2,769; Science,2016, 353,6303 and Nano Letter, 2014,15,365). Though this paper is scientifically valid and technically sound, I can not recommend it for publication in Nature Communications.

I also have several major concerns:

1. The IR-transparent nanoporous polyethylene (nanoPE) film was used as the substrate for constructing metallized nanophotonic textile in this paper, aim to keep warm by decreasing the outer surface emissivity. However, the nano PE film was reported to be a superior textile for radiative human body cooling. I see a contradiction in authors' papers. Is the nanoPE film, with metallized layer, an ideal material for "human body heating"? Did the authors try other polymers with similar structures? How about porous reflecting textiles?
2. Regarding the fabrication process, prior to the electroless plating, the author treated the nanoPE with PDA. My question is how did they avoid plugging the nanopores of the nanoPE? The laminating process can also decrease the breathability of the textiles. The detailed investigation is required.
3. In the coloration test (Figure 4f), the infrared emissivity increased by three times, which is not a "smaller increase" as the authors claimed. I doubt the feasibility of coloration.
4. Why didn't the authors consider the air and wave gaps existed in the porous materials in the heat transfer model?

Reviewer #2 (Remarks to the Author):

When heating or cooling a person by heating/cooling the space surrounding it, lots of energy is wasted for the spacious surrounding. There is a significant benefit to heat or cool locally, as tightly as a small cubic or just a human body. The energy saving potential comes from the HVAC temperature set point offset.

With a solid understanding of heat transfer processes from human body to the environment, Cui et al came up with an elegant design of new PE textile to potentially offset the space heating temperature set point. Such a design requires breathability, high IR reflectivity at the inner surface (facing human body), and low IR emissivity at the outer layer. By taking advantage of the self-generating body heat, such a novel textile could result in a temperature rise when used to cover a person. Based on the authors' recent work on IR reflecting Ag-nanowire network and radiative cooling PE textile, Cui et al constructed this interesting material: an IR-reflective layer on an IR-transparent layer with embedded nanopores that are smaller than the IR wavelength but larger than the water molecule, the nanoporous metallized polyethylene textile achieves a minimal IR emissivity of 10.1% on the outer surface that effectively suppresses heat radiation loss from the clothed human body to the environment without sacrificing wearing comfort.

The authors also made commendable tests from materials characterization, to mechanical testing, wicking test, coloration, and washability tests by comparing the new materials with some of the

existing clothes materials.

However, the title "Passive Radiative Human Body Heating by ..." is a little misleading. This reviewer suggests to change it to "Warming up Human Body by.....". This is because the key technology is "reflecting" not "passive heating".

Overall, this paper should be accepted to Nature Communications. It is rather clear that such a work will generate great interests in both thermal management, materials communities and the textile communities.

Reviewer #3 (Remarks to the Author):

This is a very interesting paper that addresses an important topic – how to reduce the heating requirements for the human environment. Reductions in the ambient temperature would significantly reduce the energy required to keep homes and business warm. Overall, I support publication of this manuscript, however I would like to see the authors address a few additional questions.

1. The authors wash the low emissivity cloth once. How do the properties change after for example 10 washes? It is ok if the properties degrade, but if they do, the authors should discuss at some level how they might maintain the silver coating w/o increasing the emissivity (may be harder than it seems, since coating of the silver with a polymer would increase the emissivity).
2. Will the silver tarnish? This is related to question 1. If there is a good way to protect the silver without significantly increasing the emissivity, it probably should be briefly discussed.
3. Were there any human trials? For example, having people wear the clothes in a dark room to have a "blind" experiment? The 7 C decrease in the required temperature is impressive, but I could see the actual number being much less given heat loss from the hands/head/face...
4. How much is the improvement over a typical indoor winter garment (e.g. shirt and sweater). This could easily be added to Fig. 3b. Yes, its great people could wear only a single shirt in the winter, but to keep warm outside, most people will probably wear more anyway.
5. How strong is the effect if the cotton/PE/Ag fabric is layered under a thicker shirt (e.g. a sweater)?

I should add, the underlying science is quite sound. My concerns are entirely that the control experiments may not be fully representative of reality, and may make this technology appear better than it actually might be.

RE: Nature Communications Manuscript Revision Request

Reviewers' comments:

Reviewer #1 (Remarks to the Author):

In this paper, the authors reported the application of the nanoporous metallized PE as the outer shell to suppress radiative heat dissipation for localized personal heating. The heat transfer, IR property and wearability were fully investigated. However, the core concept, tailoring the infrared radiate property by structure design (such as altering nano pore diameter and fiber diameter) and constructing metallized reflect layer, is ordinary due to lack of novelty. As far as I know, there have been several studies published about the similar idea (ACS Photonics, 2015,2,769; Science,2016, 353,6303 and Nano Letter, 2014,15,365). Though this paper is scientifically valid and technically sound, I can not recommend it for publication in Nature Communications.

Response: We appreciate the concern that the reviewer raised, however, the reviewer may have overlooked the novel points demonstrated in the paper.

Most of the previous studies relevant to tailoring infrared radiation property of textiles, i.e., ACS photonics,2015,2,769 and Science,2016, 353,6303, were focusing on the cooling side. Compared to cooling, much larger energy savings can be expected by developing radiative heating textiles, as space heating (22.5%) holds much larger proportion of all the energy consumed in the buildings sector than space cooling (14.8%). More research effort is needed on the heating side to fully realize the energy saving potential that the radiative heating textile endows.

For heating purpose, the AgNW coating design by Nano Letter, 2014,15,365 only achieved 0.9 °C warming than normal textiles, which is still far from optimum for effective expansion of the heating set-point. Besides, some deficiencies in the wearability of AgNW-coated textile could limit its wide adoption in practice, such as potential release of AgNW in water during washing cycles, the incapability of coloration, etc.

Currently the lack of fundamental understanding of the heat transfer process as well as deficiency in structure design are the main reasons for the lack of optimal radiative heating textile either in literature or in commercial market.

Therefore, the key novelty of this paper lies in that it not only carries out the first fundamental model analysis for heating to serve as the theoretical design guidance, but also presents a new rational structure design that can solve the dilemma between optimal radiative heating performance and good wearability. First, there is a general belief that a reflective textile inner surface is required to reflect back the IR radiation from the human body to keep warm. Our

simulation result surprisingly reveals the necessity of low-IR-emissivity at the textile outer surface in suppressing the radiative heat dissipation of clothed human, which rectifies the long term intuitive belief in the need of only reflective inner surface for heating. Second, the nanoporous metalized PE textile is a novel IR reflective/IR transparent bilayer design that together gives a low IR emissivity on the outer surface to achieve optimal heating performance, following the guidance of theoretical analysis. The use of IR-transparent nanoPE is essential and novel here, as its IR transparency does not impair the low IR emissivity of the underlying metallic layer. The nanopores of the metallic layer are theoretically and experimentally tuned to give almost 100% reflectivity while maintaining good breathability. Besides, the nanoporous metalized PE design is light-weight, durable, washable and colorable. These additional benefits also make this new bilayer structure design distinctive and superior to the previously reported AgNW coating. These novel points have been mentioned in the introduction and conclusion, and elaborated in the main text throughout the paper.

I also have several major concerns:

1. The IR-transparent nanoporous polyethylene (nanoPE) film was used as the substrate for constructing metallized nanophotonic textile in this paper, aim to keep warm by decreasing the outer surface emissivity. However, the nano PE film was reported to be a superior textile for radiative human body cooling. I see a contradiction in authors' papers. Is the nanoPE film, with metallized layer, an ideal material for "human body heating"? Did the authors try other polymers with similar structures? How about porous reflecting textiles?

Response: It is not contradictory between using nanoPE itself as the textile for cooling and using it as the substrate for constructing metallized nanophotonic textile for heating. On one hand, due to its high IR transparency, nanoPE can fully transmit human body radiation out, thus making it a superior textile for radiative cooling. On the other hand, when constructing metallic layer on nanoPE, the high IR transparency of nanoPE combined with the high IR reflectivity of the metallic layer results in the low IR emissivity of the bilayer, as $\text{emissivity} = 100\% - \text{reflectivity} - \text{transmissivity}$. The use of IR-transparent nanoPE is essential and novel here, as its IR transparency does not impair the low IR emissivity of the underlying metallic layer.

As far as we know, most of other polymers do not have high IR transparency in the wavelength range of 7-14 μm as nanoPE does. Therefore, other polymers will not work as well as nanoPE here, which is also evidenced by the high emissivity and poor radiative heating performance measured on the polyethylene terephthalate (PET) side of Mylar blanket in the paper.

In addition to the IR-transparent nanoPE, the nanoporous reflecting metallic layer is the other important component in the bilayer structure design. Both nanoPE (IR-transparent) and metallic layer (IR-reflective) are needed to achieve the low IR emissivity. The nanopores of the metallic layer are theoretically and experimentally tuned to give almost 100% reflectivity while maintaining good breathability.

2. Regarding the fabrication process, prior to the electroless plating, the author treated the nanoPE with PDA. My question is how did they avoid plugging the nanopores of the nanoPE? The laminating process can also decrease the breathability of the textiles. The detailed investigation is required.

Response: The nanoPE surface was treated with polydopamine (PDA) coating for only 2 hours. According to the literature (*Science* 2007, 318, 426-430), 2 hour coating of polydopamine corresponds to a thickness of about 10 nm. Such an ultrathin thickness will not plug the nanopores of nanoPE, as the pore size of nanoPE is in the range of 50 – 1000 nm. It is also confirmed by the SEM image and water vapor transmission measurement that the pores are not plugged by polydopamine coating.

In addition, we have investigated the breathability of the laminated textile by measuring the water vapor transmission rate (Figure 4a). Based on the water vapor transmission results, the laminated cotton/Ag/PE textile has a water vapor transmission rate at $\sim 0.012 \text{ g/cm}^2\cdot\text{hr}$, which is only slightly lower than that of cotton and Omni-Heat ($\sim 0.015 \text{ g/cm}^2\cdot\text{hr}$, Figure 4a). This results suggest that the laminating process does not considerably affect the breathability of the textiles.

3. In the coloration test (Figure 4f), the infrared emissivity increased by three times, which is not a “smaller increase” as the authors claimed. I doubt the feasibility of coloration.

Response: We tested the coloration of cotton/Ag/PE textile with both Prussian blue and permanent marker dye. The measured IR emissivities for non-colored, Prussian blue colored, marker dye colored textiles are 12.0%, 16.4% and 27.9%, respectively. The IR emissivity was not increased by three times after coloration. Instead, it was found that low-IR-absorbing pigment like Prussian blue results in smaller increase of IR emissivity of the textile (16.4% vs. 12.0%) than high-IR-absorbing pigment like permanent marker dye (27.9%). Therefore, we suggested in the manuscript that it is critical to choose pigments with low IR absorption, like Prussian blue, for coloring the radiative heating textiles. With the right choice of low-IR-absorbing pigment, it is feasible to obtain coloration on the radiative heating textile without significantly affecting its thermal property.

4. Why didn't the authors consider the air and wave gaps existed in the porous materials in the heat transfer model?

Response: The goal of the heat transfer model analysis is to provide a general theoretical guidance for the effect of IR properties on the heating performance of textiles. In the heat transfer model, the apparent parameters of the textile include its IR properties (emissivity, reflectivity and transmissivity), thermal conductivity and thickness. These apparent parameters directly determine the model analysis results. The air gap in the porous materials is an implicit factor that may affect the values of these apparent parameters (it is unclear what the wave gap that the reviewer refers to). Therefore, we only need to consider the overall values of these

apparent parameters in the model analysis, without the need to dig into the details of the implicit factors. For example, we varied the IR emissivity from 0 to 1, in order to obtain the general correlation between the IR properties and the heating performance. We also assumed a typical value of $0.05 \text{ W}\cdot\text{m}^{-1}\cdot\text{K}^{-1}$ for the thermal conductivity of the textile layer. This value considered the existence of air gap in typical textiles. On the other hand, when designing the details of the nanophotonic structure textile, we considered the air gap in our *optical structure calculations*.

Reviewer #2 (Remarks to the Author):

When heating or cooling a person by heating/cooling the space surrounding it, lots of energy is wasted for the spacious surrounding. There is a significant benefit to heat or cool locally, as tightly as a small cubic or just a human body. The energy saving potential comes from the HVAC temperature set point offset.

With a solid understanding of heat transfer processes from human body to the environment, Cui et al came up with an elegant design of new PE textile to potentially offset the space heating temperature set point. Such a design requires breathability, high IR reflectivity at the inner surface (facing human body), and low IR emissivity at the outer layer. By taking advantage of the self-generating body heat, such a novel textile could result in a temperature rise when used to cover a person. Based on the authors' recent work on IR reflecting Ag-nanowire network and radiative cooling PE textile, Cui et al constructed this interesting material: an IR-reflective layer on an IR-transparent layer with embedded nanopores that are smaller than the IR wavelength but larger than the water molecule, the nanoporous metallized polyethylene textile achieves a minimal IR emissivity of 10.1% on the outer surface that effectively suppresses heat radiation loss from the clothed human body to the environment without sacrificing wearing comfort.

The authors also made commendable tests from materials characterization, to mechanical testing, wicking test, coloration, and washability tests by comparing the new materials with some of the existing clothes materials.

However, the title "Passive Radiative Human Body Heating by ..." is a little misleading. This reviewer suggests to change it to "Warming up Human Body by.....". This is because the key technology is "reflecting" not "passive heating".

Overall, this paper should be accepted to Nature Communications. It is rather clear that such a work will generate great interests in both thermal management, materials communities and the textile communities.

Response: We appreciate the reviewer's commendatory comments and the suggestion on the title.

Revision: We have changed the title to "Warming up Human Body by Nanoporous Metallized Polyethylene Textile".

Reviewer #3 (Remarks to the Author):

This is a very interesting paper that addresses an important topic – how to reduce the heating requirements for the human environment. Reductions in the ambient temperature would significantly reduce the energy required to keep homes and business warm. Overall, I support publication of this manuscript, however I would like to see the authors address a few additional questions.

1. The authors wash the low emissivity cloth once. How do the properties change after for example 10 washes? It is ok if the properties degrade, but if they do, the authors should discuss at some level how they might maintain the silver coating w/o increasing the emissivity (may be harder than it seems, since coating of the silver with a polymer would increase the emissivity).

Response: We have carried out additional experiments to test the IR property change after 10 washes. The FTIR characterization results show that the IR emissivity of cotton/Ag/PE textile slightly increases from 16.8% to 20.0% after 10 washes (Figure 4e). Such slight increase suggests that the low-IR-emissivity textile can sustain multiple washing without significant degradation of its IR property. This is because Ag is strongly bonded to nanoPE through polydopamine coating and electroless plating process, which cannot even be removed by tape exfoliation as tested and shown in Figure 4d. Further, to better protect the silver coating against washing or tarnish, additional polymer coating can be applied as a protection layer. Note that the polymer layer is essentially added in between cotton and Ag in the **cotton/Ag/PE** textile where cotton is facing the skin and PE is facing outside environment, resulting in a structure layout in the series of **cotton/polymer/Ag/PE**. Here, the polymer coating will only increase the IR emissivity of the inner surface (facing the skin), while the outer surface (facing environment) emissivity will still maintain low. Our heat transfer model analysis demonstrates that the suppression of radiative heat dissipation of clothed human is determined by the low-IR-emissivity at the textile outer surface. Therefore, polymer coating will not decrease the performance of the textile.

Revision: We have added the IR emissivity result after 10 washes to Figure 4e.

2. Will the silver tarnish? This is related to question 1. If there is a good way to protect the silver without significantly increasing the emissivity, it probably should be briefly discussed.

Response: We have measured the IR emissivity after a duration of 10 month. The results show that the IR emissivity of cotton/Ag/PE textile slightly increases from 11.4% (0 month) to 16.8% (10 month). This suggests that silver is quite stable without significant tarnish issue. As mentioned in the response to Reviewer 3's comment 1, we can further protect the silver by adding additional polymer coating without significantly increasing the emissivity.

Revision: We have added the IR emissivity result after 10 month to Figure 4e.

3. Were there any human trials? For example, having people wear the clothes in a dark room to have a "blind" experiment? The 7 C decrease in the required temperature is impressive, but I could see the actual number being much less given heat loss from the hands/head/face...

Response: We appreciate the reviewer's suggestion of human test. We agree that the uncovered hands/head/face will result in some amount of heat loss. We did some preliminary test to have several people wearing the cloth in a cool environment (15 °C). Most of the people felt that the low-IR-emissivity textile is warmer than normal shirt under such cool environment. While the human body feeling is subjective and thermophysiology of the human body is complicated, we will need systematic and quantitative study in the future to test the thermal effect of the textile in the human scale with the consideration of those factors that the reviewer suggested.

4. How much is the improvement over a typical indoor winter garment (e.g. shirt and sweater). This could easily be added to Fig. 3b. Yes, its great people could wear only a single shirt in the winter, but to keep warm outside, most people will probably wear more anyway.

Response: We have performed additional thermal measurement of a typical indoor winter garment, *i.e.*, a shirt plus a sweater. The measured set-point requirement for shirt/sweater is 19.6 °C, which is much higher than that of our cotton/Ag/PE textile. In principle, people can wear this low-IR-emissivity shirt both inside and outside, if the ambient temperature is around 15 °C.

Revision: The thermal performance of a shirt plus a sweater is added to Figure 3b. In the "Passive radiative heating performance section" on page 8, we added the following sentence after the sentence "...which is 7.1 °C lower than that of cotton (22.1 °C)."

"The thermal performance of the cotton/Ag/PE textile is also better than typical indoor winter garments, e.g., a shirt and a sweater, which requires a set-point of 19.6 °C."

5. How strong is the effect if the cotton/PE/Ag fabric is layered under a thicker shirt (e.g. a sweater)?

Response: Based on the heat transfer model analysis, the suppression effect of radiation heat loss will be small if the emissivity of the textile outer surface is high. Therefore, theoretically, the effect will be minimal if the cotton/Ag/PE textile is layered under a sweater. We have also experimentally measured the thermal performance of cotton/Ag/PE/sweater (Figure 3b). The

measured set-point requirement for cotton/Ag/PE/sweater is 18.6 °C, which is indeed not better than cotton/Ag/PE only. However, the thicker shirt would help reducing thermal conduction.

I should add, the underlying science is quite sound. My concerns are entirely that the control experiments may not be fully representative of reality, and may make this technology appear better than it actually might be.

Response: We appreciate the concern that the reviewer brought up. We note that the goal of the paper is to provide better understanding of the fundamental science regarding the heat transfer process of the clothed human body and to develop a new textile design for optimal radiative heating. We believe this pioneer study will have great potential to offer heating energy savings of our society, and stimulate more research in many relevant areas. While the research in this area is at its infancy, we also agree that to further use this technology in real practice, a systematic and quantitative study in the human scale is needed in the future to test the thermal effect of the textile with the consideration of thermophysiology of the whole human body.

Reviewers' comments:

Reviewer #1 (Remarks to the Author):

In this paper, the IR reflective/IR transparent bilayer design with nanoporous metalized PE textile is reported. The authors have responded partial my concerns. For publication and improving the scientific significance, however, the authors should address a few additional questions.

1. For comparing with the nanoporous polyethylene textile, the author claimed that the normal textile is not satisfies IR transparency in the Schematics Figure 1B in their previous work. For comparing with the nanoporous metallized polyethylene textile in this paper, however, the authors claimed that the normal textile has strong IR emission (Thermal radiation). Cooling and warming process all are based on normal textiles, I don't know why they claimed the normal textile have the different IR transparency in their comparison.

2. The authors built the heat transfer model to analysis the effect of IR properties on the heating performance of textiles, expecting to provide a general theoretical guidance for structure design of the textiles. However, for analysing the heat transfer model, the authors ignored the effect of the air existed in the nanoporous materials and the interlayers. It may be a misleading result.

Actually, not only the IR properties (emissivity, reflectivity, and transmissivity), thermal conductivity, and thickness critical to the model and results, but also the air on no account can be ignored due to that it has decisive influence to the IR performance, thermal conductivity, and the results. Therefore, for improving the scientific significance of this paper, the authors should introduce the parameter of air effect in the heat transfer model.

Reviewer #3 (Remarks to the Author):

I am quite pleased with the revisions of the referees, in particular the inclusion of a number of control experiments. I encourage the authors to carry out human trials and report them in a subsequent publication - after all, human trials are the only way to know if the technology will truly have impact. The underlying science in the manuscript is sound, the idea is novel, and the work well performed. As such I recommend accepting the manuscript.

My only suggested revision is in Fig. 4e to not connect the dots between the time series of data and the number of washes. Perhaps have before wash, after wash, and after 10 washes connected, and then a break in the data, then 0, 1, 2, 3, 10 month data connected.

Reviewers' Comments:

Reviewer #1 (Remarks to the Author):

In this paper, the IR reflective/IR transparent bilayer design with nanoporous metallized PE textile is reported. The authors have responded partial my concerns. For publication and improving the scientific significance, however, the authors should address a few additional questions.

1. For comparing with the nanoporous polyethylene textile, the author claimed that the normal textile is not satisfies IR transparency in the Schematics Figure 1B in their previous work. For comparing with the nanoporous metallized polyethylene textile in this paper, however, the authors claimed that the normal textile has strong IR emission (Thermal radiation). Cooling and warming process all are based on normal textiles, I don't know why they claimed the normal textile have the different IR transparency in their comparison.

Response: We did not claim that normal textile has different IR transparency in the comparison. For warming or cooling purpose, the requirements of the IR property are different. To highlight the importance of the respective IR property for warming or cooling, the schematics in this paper (for warming) and the previous paper (for cooling) have different emphasis and meaning. The schematic in this paper emphasizes the relative **warming** effect of **low-IR-emissivity** textile with respect to normal textiles (high-IR-emissivity). The schematic in Figure 1B in the previous work means that normal textiles with high-IR-emissivity but low-IR-transparency cannot transmit out all the human body radiation as efficiently as **IR-transparent** textile for **cooling** purpose. The IR properties of normal textile, nanoPE (for cooling), and nanoporous metallized polyethylene (for warming) are specified and compared in the table below. Essentially, the IR radiation property of normal textile is not optimal for either warming or cooling, as it is neither low-IR-emissive nor IR-transparent.

	IR transmissivity τ (%)	IR reflectivity ρ (%)	IR emissivity ε (%)
Normal textile (e.g. , cotton)	2	9	89
NanoPE	96	3	1
Nanoporous metallized polyethylene	0	90	10

Revision: We have added the above table as Supplementary Table 1 to the Supplementary Information. In line 115 on page 5 in the main text, Supplementary Table 1 has been referred to specify the IR property of normal textile in the sentence “Normal textiles like cotton (Fig. 1b and Supplementary Table 1), however, lack the desired radiation control due to their high emissivity of 0.8 ~ 0.9...”.

2. The authors built the heat transfer model to analysis the effect of IR properties on the heating performance of textiles, expecting to provide a general theoretical guidance for structure design of the textiles. However, for analysing the heat transfer model, the authors ignored the effect of the air existed in the nanoporous materials and the interlayers. It may be a misleading result. Actually, not only the IR properties (emissivity, reflectivity, and transmissivity), thermal conductivity, and thickness critical to the model and results, but also the air on no account can be ignored due to that it has decisive influence to the IR performance, thermal conductivity, and the results. Therefore, for improving the scientific significance of this paper, the authors should introduce the parameter of air effect in the heat transfer model.

Response: The schematic of the heat transfer model is shown below (more details about the calculation are given in the Supplementary Information). Regarding the air gap involved in this system, we should note the following points:

First, we have considered the air gap between the skin and the textile in the model. The thermal conduction in the air gap is calculated as $q_{cond,a} = \frac{k_a}{t_a} (T_s - T_i)$, where k_a is the thermal conductivity of air, t_a is the air gap thickness, T_s is the skin temperature and T_i is the temperature of the textile inner surface.

Second, we consider the porous textile as an entirety in the model for calculating the thermal conduction within the textile, $q_{cond,t} = \frac{k_t}{t_t} (T_i - T_o)$, where k_t is the effective thermal conductivity of the textile, t_t is the textile thickness, T_i is the temperature of the textile inner surface and T_o is the temperature of the textile outer surface. The use of effective thermal conductivity value for the textile layer as a whole is based on effective medium theory,¹⁻⁴ which averages the multiple values

of the constituents, instead of separately considering the air pore constituent and the fiber constituent. Such effective medium approximation is often used and found acceptable in the literature to describe the parameters and properties of composite materials, as precise calculation of the many constituent values is nearly impossible.⁵⁻⁸ The effective textile thermal conductivity value of $0.05 \text{ W}\cdot\text{m}^{-1}\cdot\text{K}^{-1}$ used in our calculation is consistent with the typical values for textiles from both experimental measurement and theoretical calculation found in the literature.⁹⁻¹² Therefore, our calculation results should give reasonable estimation of heat transfer through the textile layer.

Third, due to the small size and tortuosity of the pores in the textile as well as the sedentary nature in the indoor environment, convection within the textile and air gap layer is negligible and therefore not considered. This approximation is also used and found reasonable in the literature.^{13,14}

Finally, the calculated set-point values based on the heat transfer model are in good agreement with the experimentally measured data, which further confirms that our heat transfer model is reasonably accurate.

Revision: In the supplementary information, we have added the following sentence, “*Note that the textile thermal conductivity k_{textile} is an effective value of the textile as an entirety, in which the existence of air gaps in the textile is considered.*”⁶⁻⁹. In addition, we modified the schematic of the heat transfer model in Figure S1 to better illustrate that the model includes the consideration of air gaps in the textile.

Reviewer #3 (Remarks to the Author):

I am quite pleased with the revisions of the referees, in particular the inclusion of a number of control experiments. I encourage the authors to carry out human trials and report them in a subsequent publication - after all, human trials are the only way to know if the technology will truly have impact. The underlying science in the manuscript is sound, the idea is novel, and the work well performed. As such I recommend accepting the manuscript.

My only suggested revision is in Fig. 4e to not connect the dots between the time series of data and the number of washes. Perhaps have before wash, after wash, and after 10 washes connected, and then a break in the data, then 0, 1, 2, 3, 10 month data connected.

Response and revision: We thank the reviewer for the constructive suggestion and the recommendation of publication. We have revised Figure 4e following the reviewer’s suggestion.

References:

- 1 Progelhof, R. C., Throne, J. L. & Ruetsch, R. R. Methods for predicting the thermal conductivity of composite systems: A review. *Polymer Engineering & Science* **16**, 615-625 (1976).
- 2 Choy, T. C. *Effective Medium Theory: Principles and Applications*. (OUP Oxford, 2015).
- 3 Woodside, W. CALCULATION OF THE THERMAL CONDUCTIVITY OF POROUS MEDIA. *Canadian Journal of Physics* **36**, 815-823 (1958).
- 4 Nozad, I., Carbonell, R. G. & Whitaker, S. Heat conduction in multiphase systems—I. *Chemical Engineering Science* **40**, 843-855 (1985).
- 5 Gong, L., Wang, Y., Cheng, X., Zhang, R. & Zhang, H. A novel effective medium theory for modelling the thermal conductivity of porous materials. *International Journal of Heat and Mass Transfer* **68**, 295-298 (2014).
- 6 Wang, J., Carson, J. K., North, M. F. & Cleland, D. J. A new structural model of effective thermal conductivity for heterogeneous materials with co-continuous phases. *International Journal of Heat and Mass Transfer* **51**, 2389-2397 (2008).
- 7 Zhu, F., Cui, S. & Gu, B. Fractal analysis for effective thermal conductivity of random fibrous porous materials. *Physics Letters A* **374**, 4411-4414 (2010).
- 8 Bauer, T. H. A general analytical approach toward the thermal conductivity of porous media. *International Journal of Heat and Mass Transfer* **36**, 4181-4191 (1993).
- 9 Zhu, F. & Li, K. Determining Effective Thermal Conductivity of Fabrics by Using Fractal Method. *International Journal of Thermophysics* **31**, 612-619 (2010).
- 10 Majumdar, A., Mukhopadhyay, S. & Yadav, R. Thermal properties of knitted fabrics made from cotton and regenerated bamboo cellulosic fibres. *International Journal of Thermal Sciences* **49**, 2042-2048 (2010).
- 11 Oğlakcioğlu, N. & Marmarali, A. Thermal comfort properties of some knitted structures. *Fibres & Textiles in Eastern Europe* **15**, 64-65 (2007).
- 12 Yamashita, Y., Yamada, H. & Miyake, H. Effective Thermal Conductivity of Plain Weave Fabric and its Composite Material Made from High Strength Fibers. *Journal of Textile Engineering* **54**, 111-119 (2008).
- 13 Tong, J. K. *et al.* Infrared-Transparent Visible-Opaque Fabrics for Wearable Personal Thermal Management. *ACS Photonics* **2**, 769-778 (2015).
- 14 Hsu, P.-C. *et al.* Radiative human body cooling by nanoporous polyethylene textile. *Science* **353**, 1019-1023 (2016).

REVIEWERS' COMMENTS:

Reviewer #1 (Remarks to the Author):

In this version, the authors tried their best to respond to reviewers' comments, and they have fully answered my concerns and comments. I have no more concern for the publication of this work. Therefore, I agree to recommend this paper for publication in Nature Communications.